# Petrogenesis and Tectonic Setting of the Madeng Dacite, SW Sanjiang Indosinian Orogen: Evidence from Zircon U-Pb-Hf Isotopes, and Whole-Rock Geochemistry and Sr-Nd Isotopes

**Gang Huang** [1] , **Liang-Liang Zhuang** [1,*], **Ya-Qi Yang** [2], **Li-Dan Tian** [1], **Wei Wu** [1] **and Jin-Hong Liu** [3]

[1] Institute of Geology, Chinese Academy of Geological Sciences, Beijing 100037, China; ghuang13@163.com (G.H.); tianlidan65@outlook.com (L.-D.T.); wuwei15@126.com (W.W.)

[2] School of Resources and Environment Engineering, Guizhou University, Guiyang 550025, China; yangyaqi18@163.com

[3] Xianyang Northwest Nonferrous 712 Co., Ltd., Xianyang 712000, China; liujinhong0324@163.com

[*] Correspondence: zhuangliangliang@cags.ac.cn; Tel.: +86-137-1792-8605

**Abstract:** The Sanjiang Indosinian orogen, located in the eastern part of the Paleo-Tethys tectonic domain, is a critical region to study the Paleo-Tethyan Ocean evolution. Middle Permian–Late Triassic magmatic rocks are widespread in the Deqin–Weixi–Madeng area of southwestern (SW) Sanjiang Indosinian orogen, yet their petrogenesis and tectonic setting remain disputed. In this study, LA-ICP-MS zircon U-Pb age and Hf isotopes, and whole-rock elemental and Sr-Nd isotope geochemistry of Madeng dacite were studied. The Madeng dacite was dated at ca. 241.7 and 243.4 Ma. The samples had high $Al_2O_3$ (12.91 to 14.39 wt.%) but low MgO (0.62 to 1.76 wt.%) contents, and were alkali-rich ($Na_2O + K_2O$ = 6.97 to 8.66 wt.%) with A/CNK > 1.1, strongly resembling peraluminous S-type granites. The rocks were enriched in Rb, K, Th, U and LREE, but depleted in Ba, Sr, Nb, Ta, P and Ti, and showed obvious negative Eu anomalies, suggesting fractionation of Ti-bearing minerals (e.g., rutile and ilmenite) and plagioclase. The dacite had an initial $^{87}Sr/^{86}Sr$ value of 0.705698 to 0.710277, and negative $\varepsilon_{Nd}(t)$ (−11.28 to −10.64) and $\varepsilon_{Hf}(t)$ (−13.99 to −8.60), indicating a continental meta-sedimentary source. Their average Nb/Ta (12.24) and Th/U (4.65) were also consistent with continental crust. According to the lithological assemblage and geochemical features, we propose that the Deqin–Weixi–Madeng area intermediate-felsic magmatism was generated in a subduction-related tectonic setting.

**Keywords:** Yunnan (SW China); Sanjiang Indosinian orogen; dacite; Triassic; Paleo-Tethyan subduction

## 1. Introduction

The Sanjiang Indosinian orogen is located in the southeastern margin of the Tibetan Plateau (Figure 1A,B). The orogen comprises a series of NS-/NW-trending Paleozoic suture, island arc, and micro-continental slivers [1,2]. The major sutures include the Jinshajiang–Ailaoshan and Longmucuo–Shuanghu–Changning–Menglian. This orogen has undergone multi-stage tectono-magmat activity, and recorded the complete history of the opening and closure of the Paleo-Tethys Ocean [3–7]. Situated in the northern-central part of Sanjiang orogen, the Deqin–Weixi–Madeng area is the largest magmatic belt in the orogen, which is an ideal region for understanding and reconstructing the subduction and collision process following the closure of the Paleo-Tethys.

Recently, many geochronological studies on the Deqin–Weixi–Madeng magmatic rocks have indicated the presence of Early Permian–Late Triassic magmatism, which formed a series of basaltic to rhyolitic volcanics and coeval intermediate-felsic plutons. However, the petrogenesis and tectonic setting of these magmatic rocks remain controversial, which constrains understanding of the Paleo-Tethys evolution. Some workers suggested that these volcano-plutonic assemblages were generated by the west-dipping Jinshajiang Ocean

subduction and subsequent collision and post-collision extension [8–10]. Other studies have suggested that these magmatic rocks were produced by the north-dipping subduction of the Paleo-Tethyan Longmucuo-Shuanghu branch [7,11–15].

In this paper, we focus on the Deqin–Weixi–Madeng continental margin arc belt that consists of dacite, rhyolite with minor basalt and contemporary granitoid batholiths. We report an integrated study of LA-ICP-MS zircon U-Pb geochronology, whole-rock geochemistry, and Sr-Nd-Hf isotopes on the Madeng dacite from the Lanping basin, in order to constrain the petrogenesis and origin of these rocks. Combined with published studies on the Deqin–Weixi–Madeng magmatic rocks, we reconstructed the tectonic setting and provided insights into the evolution of the Paleo-Tethys.

## 2. Geological Setting

The Sanjiang Indosinian orogen magmatic rocks can be divided into three segments with distinctive magmatism history along the strike [7], including a northern segment (north of Weixi), mainly encompassing the East Qiangtang and Zhongzan blocks; a middle segment (between Weixi and Changning), with magmatic rocks developed along the margin of Lanping basin, covered with sedimentary rocks; and a southern segment (south of Changning), with magmatic rocks developed along the margin of Simao basin. The geochronological data (see Supplementary Materials Table S1) [8,10–18] of the middle segment have revealed a single major magmatism (Figure 1C), showing a distinctive magmatic evolution in contrast with several episodes of magmatism in northern and southern segments [15]. The Deqin–Weixi–Madeng magmatic rocks are mainly exposed along the western and eastern margins of the Lanping basin in the middle segment. The magmatic rocks are distributed along a narrow N–S-trending belt (300 km long), intercalated with sandstone–siltstone and limestone. Magmatism in the middle segment likely started at 220–260 Ma (peaked at 248 Ma), and different magmatic phases are separated by periods of magmatic dormancy and sedimentation [7,14,15]. Some studies suggested that the Weixi–Madeng rocks were emplaced in one short magmatic phase (7 My) [15].

The Lanping basin is the largest Meso-Cenozoic basin in the Sanjiang Indosinian orogen and hosts abundant polymetallic deposits [19,20] (Figure 2). The main rock units exposed include Permian to Neogene sedimentary rocks and minor Permian–Middle Triassic igneous rocks on the basin margin. Thick (>1.5 km) volcanic sequence is exposed in the eastern Lanping basin (in the Weixi–Madeng–Misha), consisting mainly of rhyolite, dacite, and minor basalt [7] (Figure 2). The Madeng transection in this study (from Houdian in the west to Jiangweitang–Dapingzi in the east) can be divided into two parts separated by a thrust fault (Figure 3A). The western part contains the Permian Shanlan Formation bioclastic limestone, sandstone, and schist, Upper Triassic Sanhedong Formation fine-grained limestone, Eocene Jainchuan Formation andesitic–rhyolitic volcanic breccias, and the Miocene Sanyin Formation conglomerate sandstone with gypsum veins. The eastern part contains the Middle–Lower Triassic Pantiange Formation of dacite–rhyolite with minor basalt (from whence our samples came).

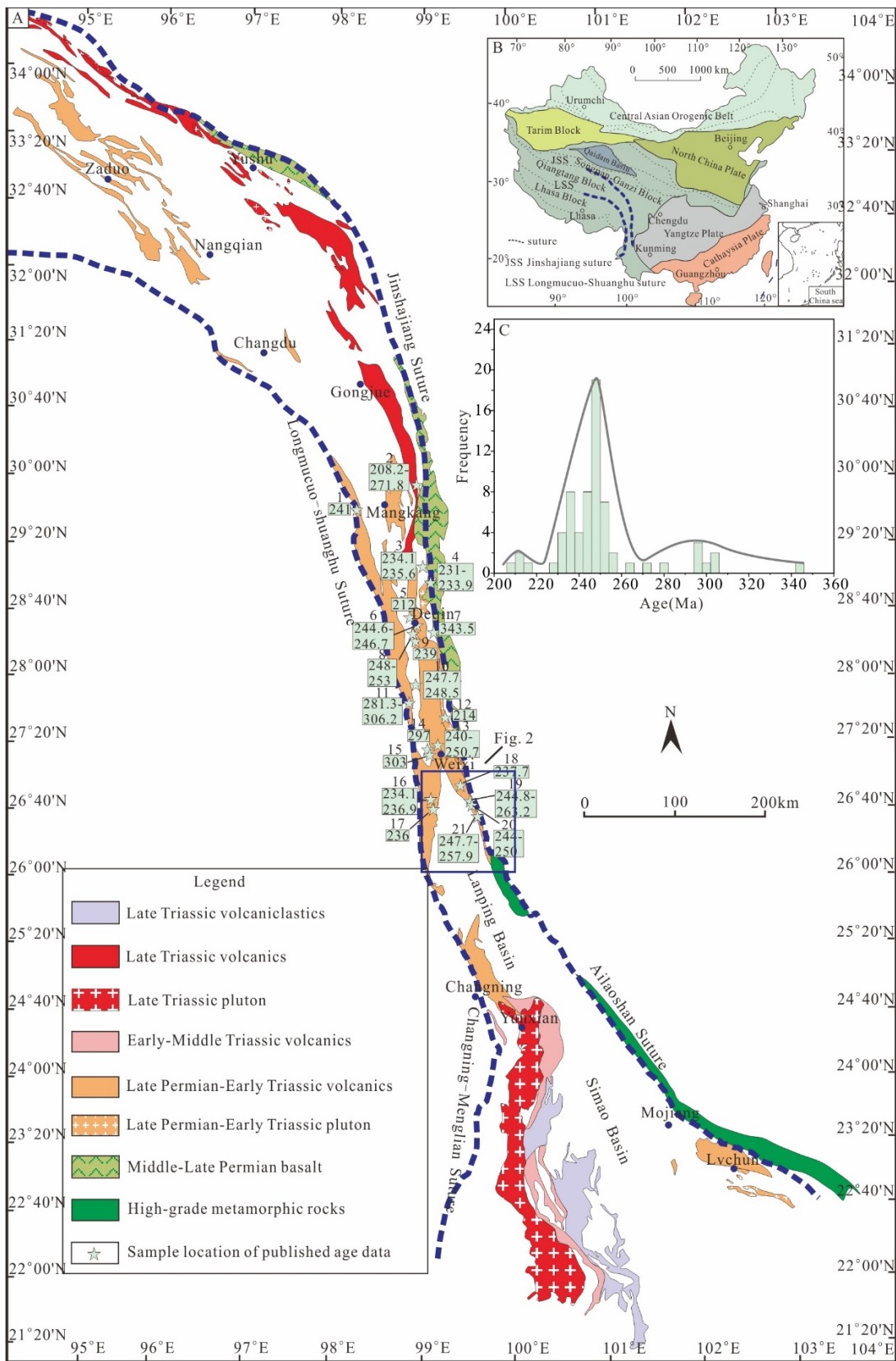

**Figure 1.** Simplified map showing the (**A**) spatial distribution of Paleo-Tethyan magmatic rocks in the Sanjiang Indosinian orogen (modified after [7]) (age data source listed in Table S1 (Supplementary Materials)), (**B**) tectonic location of the Sanjiang Indosinian orogen (modified after [21]), and (**C**) magmatic zircon age histogram for Deqin–Weixi–Madeng area.

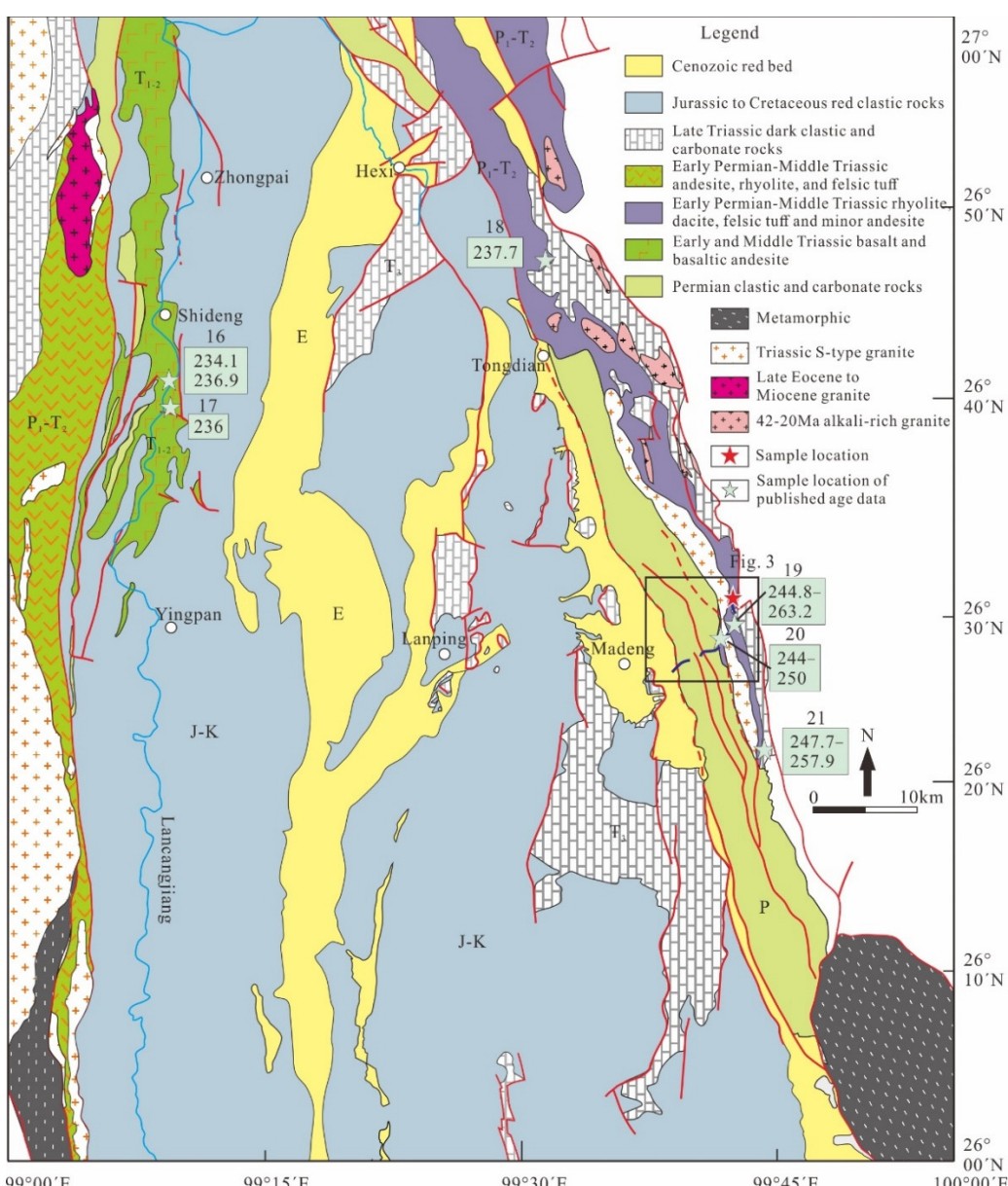

**Figure 2.** Simplified geological map of the Lanping Basin (modified after [22]), the published data source: [12–15,23].

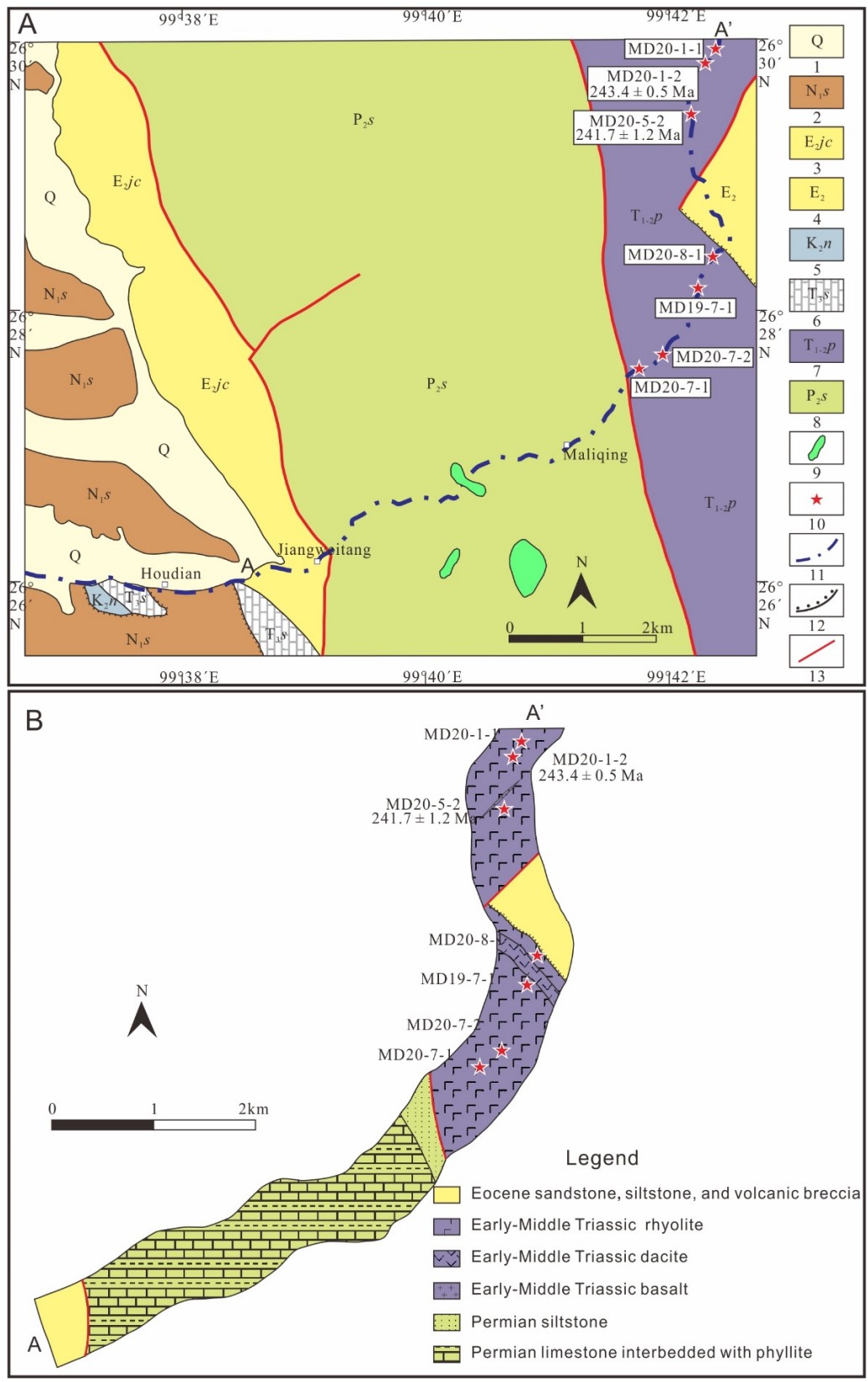

**Figure 3.** (**A**) Simplified geological map of Madeng area (modified from [14]). (**B**) Geological map of cross Section A-A' (modified after [15]). Abbreviations: 1—Quaternary; 2—Sandstone and conglomerate of the Miocene Sanying Formation; 3—Sandstone and volcanic breccia of the Eocene Jianchuan Formation; 4—Eocene volcanic breccia clast-bearing siltstone and sandstone; 5—Siltstone

and sandstone of the Cretaceous Nanxing Formation; 6—Limestone of the Late-Triassic Sanhedong Formation; 7—Dacite and rhyolite with thin basalt interlayer of the Early-Middle Triassic Pantiange Formation; 8—Limestone, phyllite and siltstone of the Permian Shanglan Formation; 9—Diabase; 10—Sample Location; 11—Road; 12—Unconformity; 13—Fault.

## 3. Sampling and Analytical Techniques

### 3.1. Sampling and Petrography

Representative, least-altered dacite samples were collected from the outcrops in Pantiange Formation at Madeng, with the sampling locations shown in Figure 3A,B.

The dacite is medium- to coarse-grained porphyritic with massive structure (Figure 4a,b). The phenocrysts include quartz, plagioclase, and feldspar, which are set in a cryptocrystalline groundmass of similar mineral content. Quartz (20%) is anhedral granular with undulose extinction. Plagioclase (25%) is subhedral–euhedral thick tabular (size: 0.5–3 mm) with polysynthetic twinning. K-feldspar (15%) is subhedral–anhedral granular with common Carlsbad twinning (Figure 4c,d).

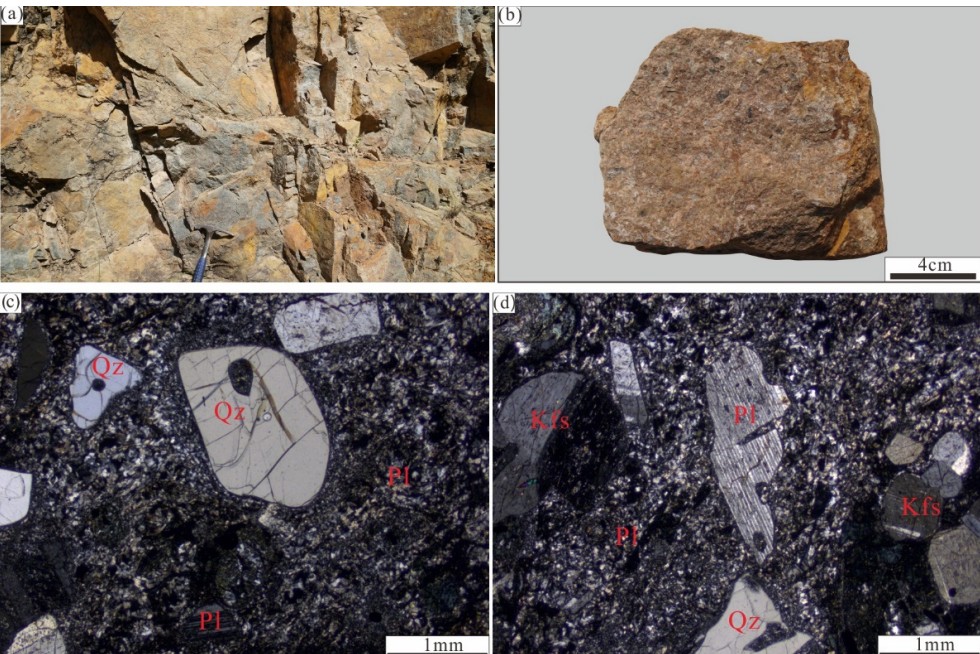

**Figure 4.** Photos/photomicrographs of the Madeng dacite: (**a**) porphyritic dacite outcrop; (**b**) dacite hand-specimen; (**c–d**) thin-section photomicrograph of dacite. Abbreviation: Qz—Quartz; Kfs—K-feldspar; Pl—Plagioclase.

Two representative dacite samples were chosen for zircon U-Pb-Hf isotope analyses. An additional seven samples were selected for major and trace element analysis and four for Sr-Nd isotope analysis. The coordinates and lithology of each sample are listed in Table 1.

**Table 1.** Summary of samples from Madeng area.

| Sample No. | Coordinates | Lithology |
|---|---|---|
| MD20-1-1 | 26°30′4.4″ N, 99°42′26.5″ E | dacite |
| MD20-1-2 | 26°30′2.5″ N, 99°42′25.5″ E | dacite |
| MD20-5-2 | 26°29′25.0″ N, 99°42′12.4″ E | dacite |
| MD20-8-1 | 26°28′22.5″ N, 99°42′21.7″ E | dacite |
| MD19-7-1 | 26°28′20.1″ N, 99°42′19.8″ E | dacite |
| MD20-7-1 | 26°27′32.1″ N, 99°41′40.8″ E | dacite |
| MD20-7-2 | 26°27′34.4″ N, 99°41′42.4″ E | dacite |

### 3.2. Analytical Methods

#### 3.2.1. LA-ICP-MS Zircon U-Pb Dating

Zircon grains were separated using conventional heavy liquid and magnetic separation techniques. Representative zircon grains were handpicked and then mounted in epoxy resin, polished, and carbon-coated. To examine the internal structure and choose the suitable analysis site, transmitted-/reflected-light micrographs and cathodoluminescence (CL) images were acquired at the Beijing Zhongke Mining Technology Co. Ltd., Beijing, China, with a MIRA3 scanning electron microscope. Euhedral–subhedral zircon grains with clear oscillatory zoning, and without fractures or inclusions, were selected for dating, using an Agilent 7770e inductively coupled plasma mass spectrometry (ICP-MS) and a MicroLas COMPexPro102 (193 nm) laser ablation system (Agilent Technologies Inc., Palo Alto, CA, USA), following the method of [24]. The data were processed with ICPMSDataCal, which was calibrated with the zircon standard 91,500 and glass NIST-610. The zircon standard GJ-1, and Plešovice yield concordia age of $339.1 \pm 1.8$ Ma and $604.0 \pm 1.9$ Ma, were consistent with their recommended values [25]. Common Pb correction was subsequently carried out with the method of [26]. The age calculation and concordia plotting were made with Isoplot 3.0 software [27]. The age uncertainty is quoted at $2\sigma$.

#### 3.2.2. Whole-Rock Geochemistry and Sr-Nd Isotopes

Fresh and clean samples were milled to ~200 mesh with an agate mill. Major and trace element analyses were carried out at the National Research Center for Geoanalysis, Chinese Academy Geological Science (Beijing, China). Major element oxides were analyzed on a PW4400 X-ray fluorescence spectrometer (XRF) with the analytical precision and accuracy being better than 5%. Trace element analysis was conducted at the same laboratory with solution ICP-MS (model PE300Q). About 50 mg of rock powder was weighted and then dissolved with $HF-HNO_3$ (1 mL: 1 mL) and digested in a Teflon bomb at 190 °C for over 24 h. The analytical precisions are better than 5% for elements at concentration >10 ppm, and less than 10% for elements with concentration <10 ppm.

Whole-rock Sr and Nd isotope analyses were performed with a Neptune Plus multi-collector (MC)-ICP-MS at the Wuhan Sample Solution Analytical Technology Co., Ltd. (Hubei, China). About 50–200 mg of sample powder was dissolved in 1–3 mL $HNO_3$ and 1–3 mL HF in a closed Teflon bomb at 190 °C for over 24 h. Subsequently, the samples were digested in 1 mL HCl. The mass fractionation corrections for $^{87}Sr/^{86}Sr$ and $^{143}Nd/^{144}Nd$ ratios were normalized to $^{86}Sr/^{88}Sr = 0.1194$ and $^{146}Nd/^{144}Nd = 0.7219$, respectively. For every seven samples, one international NBS987 and GSB standard was measured, which yielded $^{87}Sr/^{86}Sr = 0.710242 \pm 0.000012$ and $^{143}Nd/^{144}Nd = 0.512440 \pm 0.000008$, respectively, consistent (within error) with their recommended values [28,29]. The USGS BCR-2 basalt and RGM-2 rhyolite standards were chosen as the reference materials [30,31].

#### 3.2.3. LA-ICP-MS Zircon Hf Isotope Analysis

This analysis was conducted on a Neptune Plus MC-ICP-MS, equipped with a Geo-Las HD laser-ablation system at Wuhan Sample Solution Analytical Technology Co. Ltd., Wuhan, China, Helium was used as the carrier gas, with the addition of a small amount of nitrogen to improve the signal sensitivity [32]. The analysis was performed with 44 μm spot size, 10 J/cm$^2$ energy density, and 10 Hz repetition rate. The analysis was made on the same U-Pb dated zircon grains. Instrumental and data acquisition protocols were as described by [32]. During the analysis, the zircon standards of Plešovice, 91,500 and GJ-1 were analyzed, and obtained Hf isotope compositions (Plešovice = $0.282478 \pm 0.000010$, 91,500 = $0.282298 \pm 0.000011$ and GJ-1 = $0.282007 \pm 0.000010$) are consistent with the recommended value [33]. The initial $^{176}Hf/^{177}Hf$ and $\varepsilon Hf(t)$ values were calculated with reference to the present-day chondritic reservoir $^{176}Hf/^{177}Hf = 0.282785$ and $^{176}Lu/^{177}Hf = 0.0336$ at the measured U-Pb ages [34]. Single-stage Hf model ages ($T_{DM1}$) were calculated with reference to the depleted mantle at the present-day $^{176}Hf/^{177}Hf = 0.28325$ and

$^{176}$Lu/$^{177}$Hf = 0.0384 [35]. Two-stage Hf model ages (T$_{DM2}$) were calculated by assuming an average continental crust with $^{176}$Lu/$^{177}$Hf = 0.015 [36].

## 4. Results

### 4.1. Zircon U-Pb Age

Zircon grains from samples MD20-1-2 and MD20-5-2 have very similar characteristics. Most zircon grains are gray to black in CL images (Figure 5a,c). They are euhedral to subhedral and 100–300 μm in length (aspect ratios = 1:1 to 3:1). Most grains show a clear core-rim texture and well-developed oscillatory zoning in CL images (Figure 5a,c), suggesting an igneous origin [37].

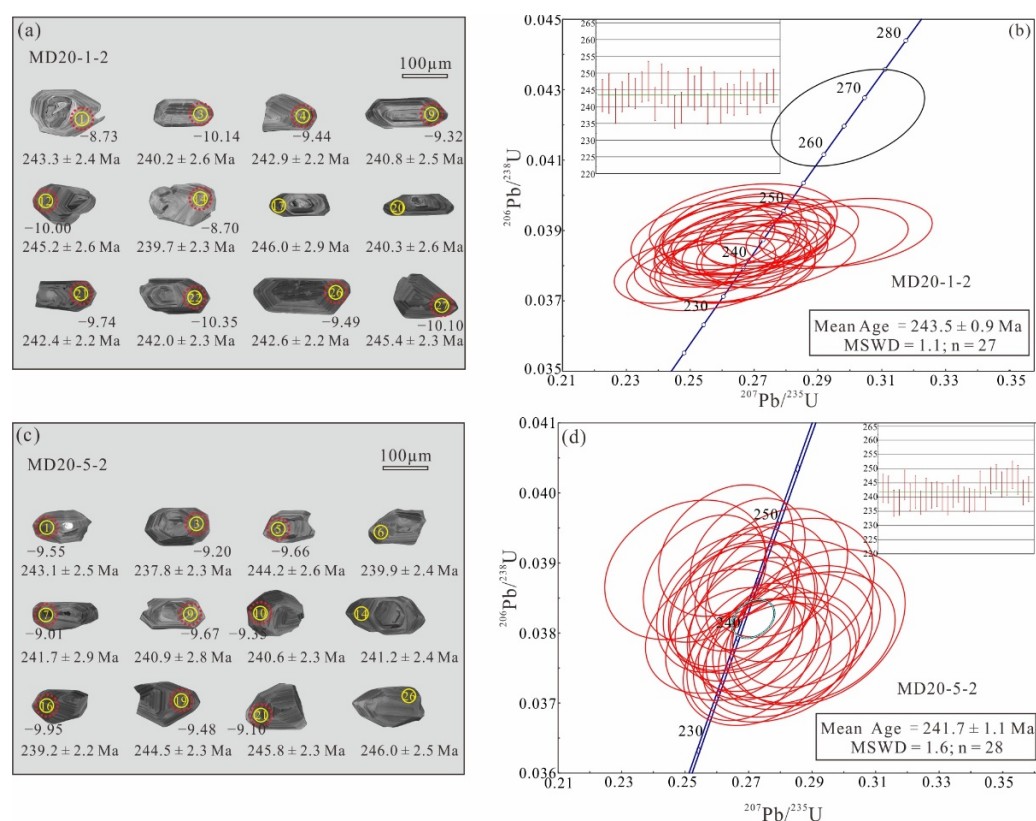

**Figure 5.** Cathodoluminescence images of representative zircon (**a**,**c**) and zircon U-Pb concordia plots (**b**,**d**) for the Madeng dacite.

Zircons from MD20-1-2 contained low and varying Th (47–168 ppm) and U (299–553 ppm) contents, with Th/U = 0.17–0.35. Twenty-eight zircons returned $^{206}$Pb/$^{238}$U ages from 238 to 266 Ma, among which 27 grains yielded a concordant weighted mean age of 243.4 ± 0.5 Ma (MSDW = 1.1, n = 27) (Figure 5a,b). Zircons from MD20-5-2 also have low and varying Th (74–363 ppm) and U (289–948 ppm) content, with Th/U = 0.2–0.56. These magmatic zircons have concordant $^{206}$Pb/$^{238}$U ages of 238 to 248 Ma with a weighted mean of 241.7 ± 1.1 Ma (MSDW = 1.6, n = 28) (Figure 5c,d). The zircon U-Pb data of our two samples indicate a Middle Triassic age of the dacite. The complete set of zircon U-Pb data is given in Table S2 (Supplementary Materials).

### 4.2. Whole-Rock Major and Trace Elemental Geochemistry

The whole-rock major and trace element results are presented in Table S3 (Supplementary Materials). The loss on ignition (LOI) for all samples below 4 wt.% suggests insignificant post-magmatic alteration or weathering. All samples exhibit similar chemical compositions, with medium SiO$_2$ (67.08–70.51 wt.%) and high Al$_2$O$_3$ (12.91–14.39 wt.%), total alkali (N$_2$O + K$_2$O = 6.97–8.66 wt.%) contents and an A/CNK

value (molar $Al_2O_3/(CaO + Na_2O + K_2O)$) of 1.14–1.57. All samples fall into the dacite field in both the TAS and $Nb/Y$-$SiO_2$ plots (Figure 6a,b) and are shoshonitic (Figure 6c). In the $A/NK$ vs. $A/CNK$ plot (Figure 6d), the samples are classified as strongly peraluminous. They also have high $FeO_T$ (2.16–5.4 wt.%) but low MgO (0.62–1.76 wt.%; Mg # = 29.39–39.29) content.

The Madeng dacite also displays similar trace element features. The dacite has high total rare earth element ($\Sigma$REE) concentrations of 164 to 274 ppm. All samples are more enriched in light REEs (LREEs) than heavy REEs (HREEs) in the chondrite-normalized REE patterns (Figure 7a) with LREE/HREE = 7.56–9.49 and $(La/Yb)_N$ = 8.56–10.65, and have distinctly negative Eu anomalies (Eu/Eu * = 0.42–0.54), indicating plagioclase fractionation. Similar to the average middle–upper continental crust [38], the Madeng dacite displayed clearly negative Ba, Sr, Nb, Ta, P and Ti anomalies, and was enriched in Th, U, Rb and K when normalized to the primitive mantle (Figure 7b).

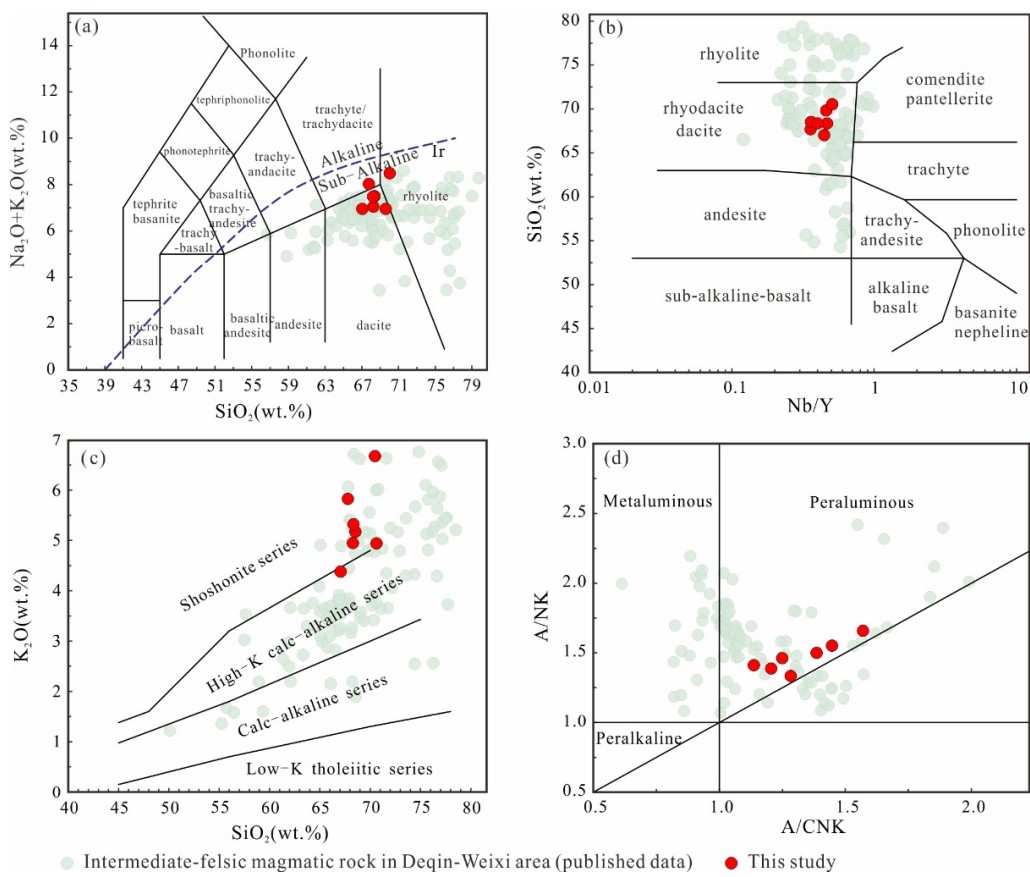

**Figure 6.** Geological discrimination plots for Deqin–Weixi intermediate-felsic magmatic rocks: (**a**) total alkali-silica (TAS) [39]; (**b**) $Nb/Y$-$SiO_2$ [40]; (**c**) $SiO_2$-$K_2O$ [41]; (**d**) $A/CNK$-$A/NK$ [42]. Data source: [8,10–12,15,43].

### 4.3. Zircon Lu-Hf Isotopic Compositions

The zircon Hf isotope data are presented in Table S4 (Supplementary Materials). Zircon grains from MD20-1-2 are relatively consistent $^{176}Hf/^{177}Hf$ from 0.282229 to 0.282385, corresponding to negative $\varepsilon_{Hf}(t)$ from −13.99 to −8.6 (n = 28) at 243.4 Ma (Figure 8a). Their two-stage model ages ($T_{DM2}$) vary from 1817 to 2157 Ma (Figure 8b). The zircon grains from sample MD20-5-2 have $^{176}Hf/^{177}Hf$ from 0.282328 to 0.282384, yielding negative $\varepsilon_{Hf}(t)$ from −10.89 to −8.72 with $T_{DM2}$ from 1821 to 1959 Ma.

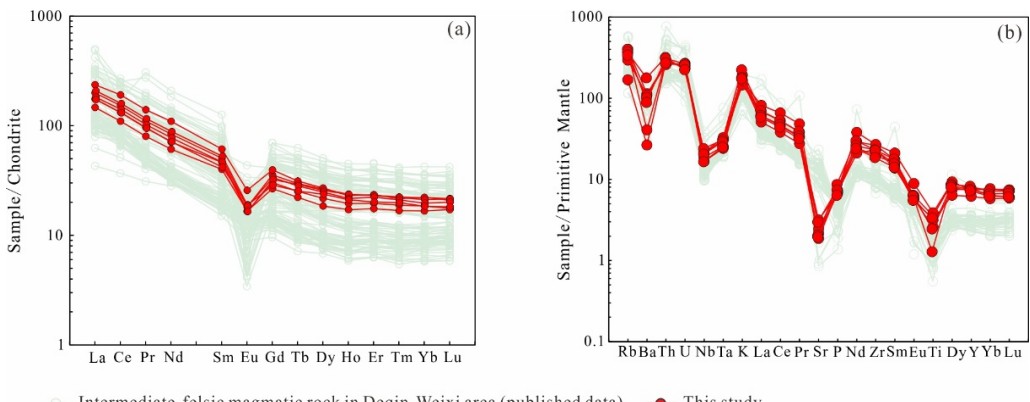

**Figure 7.** Chondrite-normalized REE patterns (**a**) and primitive mantle-normalized trace element patterns (**b**) for Deqin–Weixi intermediate-felsic magmatic rocks (normalizing values after [44]). Data source: [8,10–12,15,43].

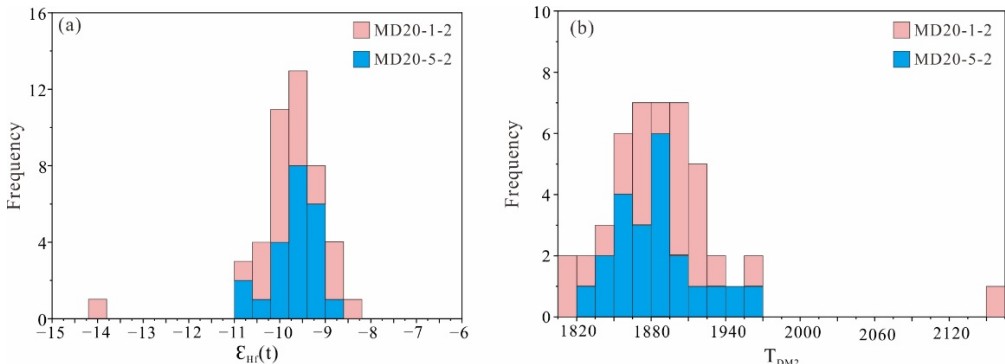

**Figure 8.** Histograms of $\varepsilon_{Hf}(t)$ (**a**) and two-stage Hf model age ($T_{DM2}$) (**b**) of the Madeng dacite.

### 4.4. Whole-Rock Sr-Nd Isotope Compositions

The Sr-Nd isotope data are presented in Table S5 (Supplementary Materials). Initial Sr isotope ratios and $\varepsilon_{Nd}(t)$ values were calculated with the obtained zircon U-Pb age of 242 Ma. The two samples have a much higher initial $^{87}Sr/^{86}Sr$ (0.738721 to 0.753399) than the present-day primitive mantle value of 0.7045 [45]. Both samples have high Rb (107–255 ppm) but relatively low Sr (39.4–66.7 ppm) contents. The $\varepsilon_{Nd}(t)$ value is negative (−11.28 to −10.64), with a $T_{DM2}$ model age of 1881 to 1933 Ma.

## 5. Discussion

### 5.1. Timing of Magmatism

We compiled 62 published geochronological data from Deqin–Weixi–Madeng area, which yielded an age from 220 to 260 Ma and represent the timing of regional magmatism (Table S1). Our new zircon ages of 243.4 and 241.7 Ma fall within this range, indicating that extensive volcanism occurred in Madeng during the Middle Triassic.

Previous geochronological studies revealed long-lived tectono-magmatic activity in the Late Paleozoic–Early Mesozoic, which formed a series of intermediate-felsic and intermediate-mafic volcanic units along the Jomda–Weixi–Yunxian continental arc (Figure 1A). Age statistics show that the magmatic rocks along the magmatic arc were accumulated from 208.2 to 343.5 Ma [8,9,12–15,43,46,47]. Combined with published data, the magmatism in Deqin–Weixi–Madeng occurred mainly from 220 to 260 Ma (peak at 248 Ma) (Figure 1C), contemporaneously with the Jomda–Weixi arc volcanism, which indicates that this magmatic arc extends from Jiangda in the north to Weixi–Madeng in the south.

### 5.2. Petrogenesis and Magma Source

#### 5.2.1. Genetic Type

The Madeng dacite is strongly peraluminous (A/CNK = 1.14–2.45), intermediate $SiO_2$ (67.08–70.51 wt.%) and high $K_2O$ (4.39–6.67 wt.%) with a relatively low differentiation index (DI) of 78.88–87.80 (Figure 5), resembling weakly fractionated S-type granites rather than I- and A-types [48,49]. Apatite has a high solubility in strongly peraluminous magma, resulting in an increase in $P_2O_5$ with increasing $SiO_2$ [50]. Therefore, the relationship between $P_2O_5$ and $SiO_2$ can be used to distinguish the magmatic affinity. The positive correlation between $P_2O_5$ and $SiO_2$ suggests that the Madeng dacite belongs to S-type granites (Figure 9a). In the 10,000 * Ga/Al diagram, all samples plot in the S-/I-type granite field with low Ga/Al ratios [51] (Figure 9b–d). In the $SiO_2$ vs. Zr, and ACF discrimination diagrams, all the Madeng dacite samples plot in the S-type granite field (Figure 9e,f), suggesting unfractionated S-type granite.

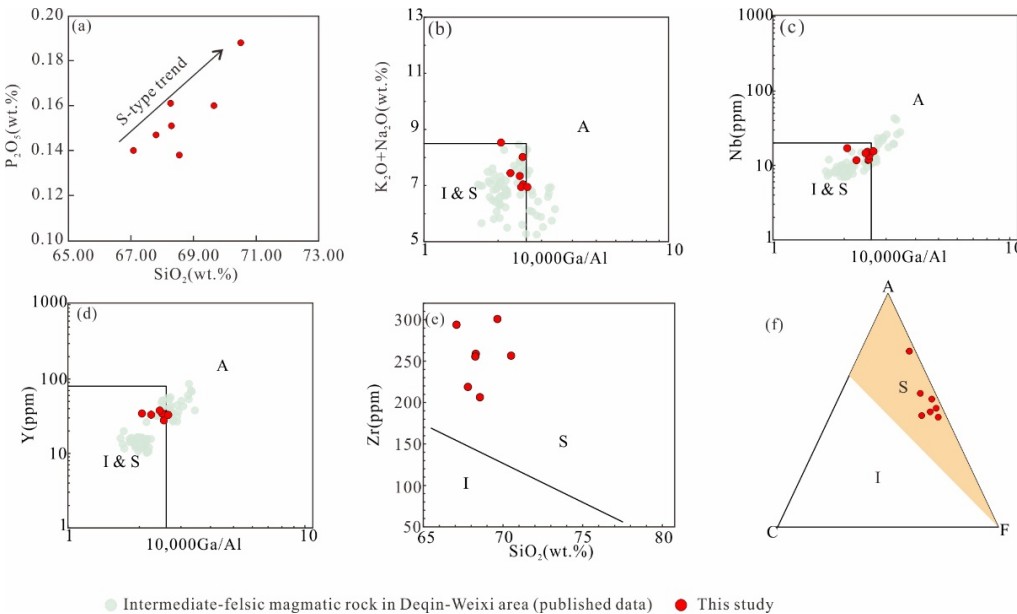

**Figure 9.** Petrogenetic discrimination plots for the Madeng dacite. (**a**) $P_2O_5$ vs. $SiO_2$; (**b–d**) 10,000 Ga/Al vs. ($K_2O + Na_2O$), Nb and Y [51]; (**e**) $SiO_2$ vs. Zr; (**f**) A($Al_3O_2$-$Na_2O$-$K_2O$)-C(CaO)-F($FeO_T$ + MgO) [50]. Data source: [8,10–12,15,43].

#### 5.2.2. Magma Source

It has been shown that some incompatible element ratios, such as Nb/Ta and Th/U, can be a useful tracing index to constrain source regions [52]. Madeng dacite has Nb/Ta = 11.41–13.62 (average 12.24) and Th/U = 3.89 to 5.73 (average 4.65), close to the average crustal values (Nb/Ta = 11–12; Th/U = 3.8–6) but distinct from the average mantle values (Nb/Ta = 17.5; Th/U = 4) [38,53]. This indicates that the Madeng dacite was derived mainly from the partial melting of the crust. In addition, Madeng dacite has negative $\varepsilon_{Hf}(t)$ = −13.99 to −8.6 with a two-stage model age of 1821 to 1959 Ma, resembling rocks derived from an ancient crustal source (Figure 10a) [54]. This conclusion is also supported by the low whole-rock $\varepsilon_{Nd}(t)$ (−11.28 to −10.64) and initial $^{87}Sr/^{86}Sr$ (0.705698 to 0.710277) (Figure 10b).

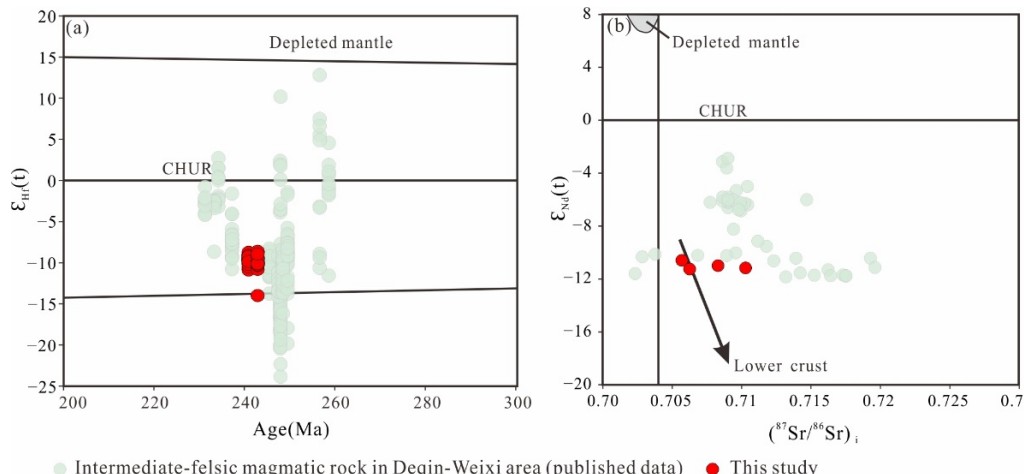

**Figure 10.** Zircon U-Pb age vs. $\varepsilon_{Hf}(t)$ (**a**) and $(^{87}Sr/^{86}Sr)_i$ vs. $\varepsilon_{Nd}(t)$ (**b**) for the Deqin–Weixi–Madeng intermediate-felsic magmatic rocks. Data source: Figure 10a [8,10,11,15]; Figure 10b [8,9,11–13].

Previous studies proposed that strongly peraluminous S-type granites are mainly derived from the partial melting of meta-sediment (e.g., meta-pelite or meta-greywacke) [49,55]. The Madeng dacite is characterized by low molar $CaO/(MgO + FeO_T)$ (0.05–0.27) and $Al_2O_3/(MgO + FeO_T)$ (1.10 to 3.10) ratios, and plots within the metapelite field (Figure 11a). In addition, experimental data have suggested that the $CaO/Na_2O$ ratio of strongly peraluminous S-type granites is mainly controlled by their source composition [56]. Melt derived from metapelite has a lower $CaO/Na_2O$ ratio (<0.5) than that derived from meta-greywacke (mainly >0.5) [55]. The Madeng dacite has a low $CaO/Na_2O$ ratio (0.13–0.39), similar to melt derived from a metapelite source. In the $CaO/Na_2O$ vs. $Al_2O_3/TiO_2$ plot, all the samples plot in the metapelite field (Figure 11b). Furthermore, in the Rb/Sr vs. Rb/Ba discrimination plot (Figure 11c), the Madeng dacite has higher Rb/Sr (2.72–5.67) and Rb/Ba (0.19–0.83) ratios, again resembling metapelite-derived melts. Therefore, the partial melting of metapelite from the ancient crust was likely the dominant mechanism of the formation mechanism for the Madeng S-type dacite.

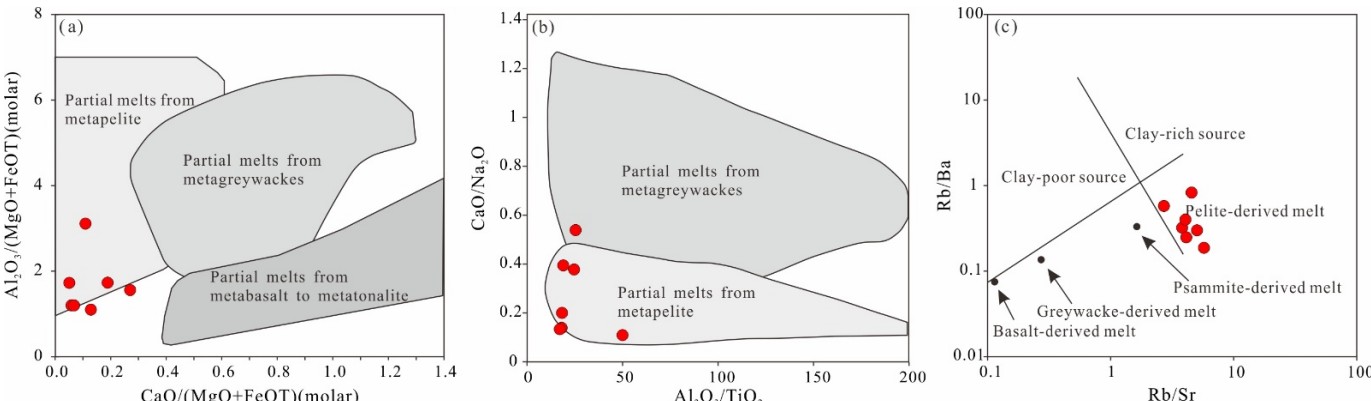

**Figure 11.** Magma source discrimination diagrams of Madeng dacite. (**a**) $CaO/(MgO + FeO_T)$ vs. $Al_2O_3/(MgO + FeO_T)$ [57]; (**b**) $Al_2O_3/TiO_2$ vs. $CaO/Na_2O$ [55]; (**c**) Rb/Sr vs. Rb/Ba [58].

### 5.2.3. Magmatic Evolution

The linear correlations between MgO, $Fe_2O_3T$, $Al_2O_3$, $TiO_2$, CaO, $Na_2O$ and $SiO_2$ for the Deqin–Weixi–Madeng magmatic rocks indicate that fractionation occurred (Figure 12). The decreasing MgO, $Fe_2O_3T$ and $Al_2O_3$ with $SiO_2$ suggest the fractionation of mafic minerals (e.g., amphibole), whilst the decreasing $TiO_2$ with $SiO_2$ indicates the fractionation of Ti oxides (e.g., rutile and ilmenite), as also supported by the distinct negative anomalies of

Ti, Nb and Ta (Figure 7b). Moreover, the negative correlation between CaO and Na$_2$O with SiO$_2$, and the distinct negative Eu, Sr and Ba anomalies suggest the significant fractionation of plagioclase. In conclusion, the partial melting of ancient crust material and fractional crystallization plays an important role in the formation of Madeng dacite.

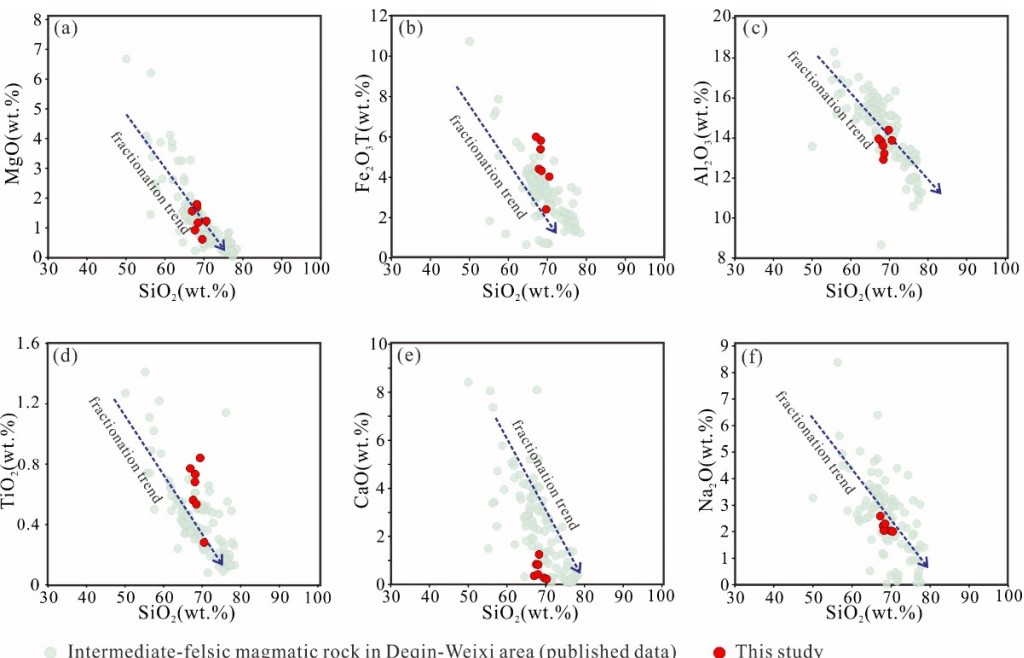

**Figure 12.** Harker diagrams for the Deqin–Weixi–Madeng intermediate-felsic magmatic rocks. (**a**) SiO$_2$ vs. MgO; (**b**) SiO$_2$ vs. Fe$_2$O$_3$$^T$; (**c**) SiO$_2$ vs. Al$_2$O$_3$; (**d**) SiO$_2$ vs. TiO$_2$; (**e**) SiO$_2$ vs. CaO; (**f**) SiO$_2$ vs. Na$_2$O. Data source: [8,10–12,15,43].

### 5.3. Tectonic Implications

Our study demonstrated that the ca.243 Ma S-type felsic rocks in Madeng were formed by partial melting of meta-pelite rocks. Previous studies have shown that S-type granites could be formed in subduction and syn-/post-collision settings [55,59,60]. As mentioned above, the Deqin–Weixi–Madeng magmatic rocks were proposed to have formed under (1) a collision and post-collision extensional [8–10] or (2) a subduction tectonic setting [7,11–15]. Some authors suggested that the Deqin–Weixi–Madeng magmatic rocks were generated during the closure of the remnant Jinshajiang Ocean, as supported by the occurrence of an ophiolite belt in the Zhongzan Block and foreland-basin molasse with bimodal volcanic rocks in the Jomda–Deqin–Weixi area. Some other authors, however, questioned the potential relationship between the ophiolite belt and remnant ocean and disputed the occurrence of the Jinshajiang suture between the Zhongzan and Changdu–Lanping–Simao Blocks. Furthermore, they proposed that regional magmatic arcs were formed by north- or east-dipping subduction of the Paleo-Tethyan Longmucuo–Shuanghu Ocean beneath the Yangtze block, based on the extensive occurrence of blueschist and eclogite in Longmucuo–Shuanghu–Changning–Menglian suture [61,62].

Our data compilation and comparison suggest that the intermediate-felsic magmatic rocks from Deqin–Weixi–Madeng have similar geochemical and Sr-Nd-Hf isotopic characteristics, and were likely formed in a similar tectonic setting and from a similar magma source. Rock assemblages in Deqin–Weixi–Madeng (basalt to rhyolite volcanic and plutonic rocks) are also typical of those occurring in the subduction zone [63]. Furthermore, the large ion lithophile element (LILE, e.g., Rb and K) enrichments and high-field strength element (HFSE, e.g., Nb, Ta, P and Ti) depletions indicate a subduction setting [64]. The low Sr/Y ratio (<20) with low Y and Yb content and low Yb$_N$ with low La$_N$/Yb$_N$ are typical of normal arc rocks (Figure 13a,b), and the Deqin–Weixi–Madeng magmatic rocks plot largely

in the volcanic arc granite field in the (Y + Nb) vs. Rb and Yb vs. Ta tectonic discrimination diagrams (Figure 13c,d). Consequently, we suggest that the Deqin–Weixi–Madeng intermediate-felsic magmatic rocks were formed in a subduction setting.

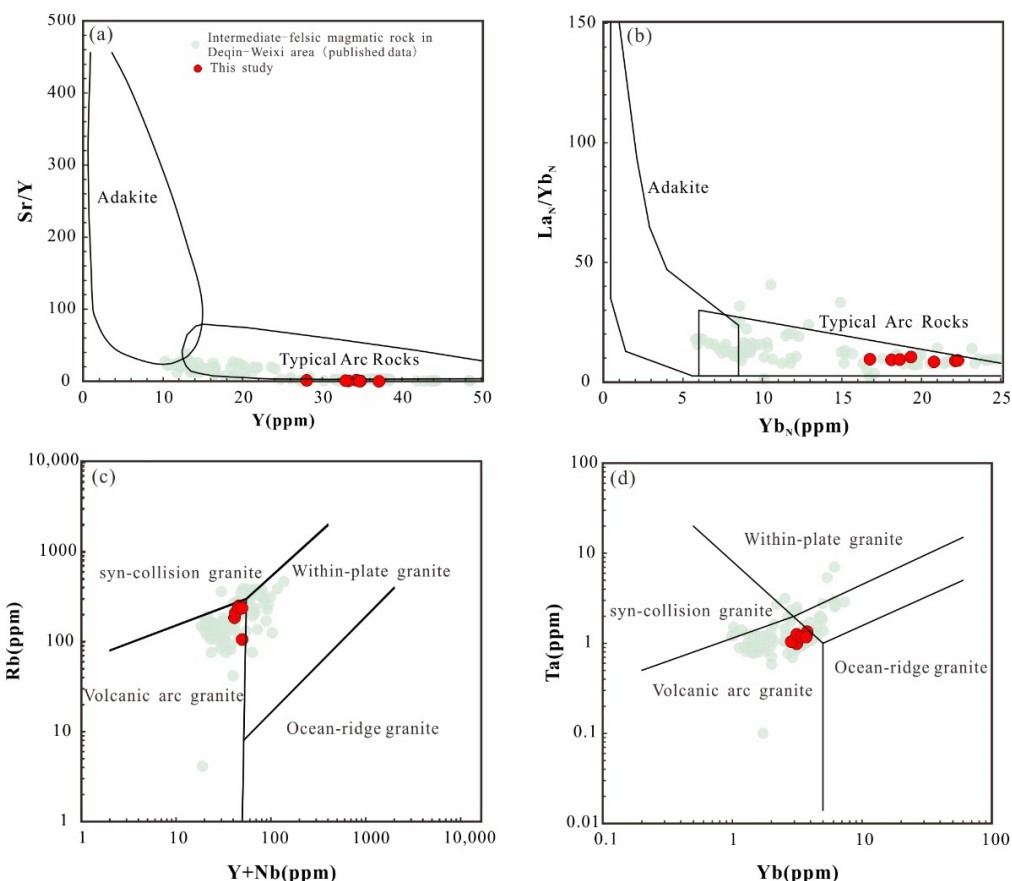

**Figure 13.** Tectonic discrimination plots for the Madeng dacite: (**a**) Y vs. Sr/Y [65]; (**b**) $Yb_N$ vs. $La_N/Yb_N$ [66] (N-Chondrite normalized [44]); (**c**) Y + Nb vs. Rb [67]; (**d**) Yb vs. Ta [67]. Data source: [8,10–12,15,43].

In brief, we suggest that the Paleo-Tethyan Longmucuo–Shuanghu Ocean may have continued to subduct during the Early Permian to Late Triassic. The long-lived dehydration of the Paleo-Tethys subducting slab and the partial melting of the mantle wedge may have generated (and accumulated) vast volumes of magma beneath the overriding plate. The crust of the overriding plate may have then been heated and partially melted to form intermediate-felsic magmas, which erupted in a short period of time (peak at 250 Ma).

## 6. Conclusions

1. LA-ICP-MS zircon U-Pb dating on the Madeng dacite yielded 241.7 and 243.4 Ma. Regional magmatic age correlation suggests that this Middle Triassic volcanism was likely formed in a subduction setting, indicating that the Paleo-Tethyan Longmucuo–Shuanghu Ocean subduction may have persisted in the Triassic.
2. The Madeng dacite is characterized by being high-Al, alkali-rich, and low-Mg. The rocks are peraluminous S-type, and display clear LILE enrichments and HFSE depletions with markedly negative Eu anomalies. This suggests the fractionation of Ti-bearing minerals (e.g., rutile and ilmenite) and plagioclase.
3. The initial $^{87}Sr/^{86}Sr$ ratios (0.705698–0.710277) with negative $\varepsilon_{Nd}(t)$ values (−11.28 to −10.64) and $\varepsilon_{Hf}(t)$ values (−13.99 to −8.6), together with the average Nb/Ta (12.24) and Th/U (4.65), indicate that Madeng dacite was formed from partial melting of the ancient crustal (meta-sedimentary rocks).

**Supplementary Materials:** The following supporting information can be downloaded at: http://www.mdpi.com/xxx/s1, Table S1: Summary of geochronological data of the Paleo-Tethyan magmatic rocks in the Deqin–Weixi–Madeng area; Table S2: Zircon U-Pb data of Madeng dacite; Table S3: The result of major oxides (wt.%) and trace elements ($\times 10^{-6}$) of Madeng dacite; Table S4: Zircon Lu-Hf isotope data of Madeng dacite; Table S5: Whole-rock Sr-Nd isotope data of Madeng dacite.

**Author Contributions:** Conceptualization, G.H. and L.-L.Z.; methodology, L.-L.Z.; software, G.H.; validation, G.H. and L.-L.Z.; formal analysis, G.H.; investigation, G.H., L.-L.Z., L.-D.T. and W.W.; resources, L.-L.Z.; data curation, G.H.; writing—original draft preparation, G.H.; writing—review and editing, L.-L.Z., Y.-Q.Y., L.-D.T., W.W. and J.-H.L.; visualization, G.H.; supervision, L.-L.Z. and Y.-Q.Y.; project administration, L.-L.Z.; funding acquisition, L.-L.Z. All authors have read and agreed to the published version of the manuscript.

**Funding:** This research was funded by the National Natural Science Foundation of China, grant number "41773043, 91855214, 41773042, 41772088 and 41922022".

**Data Availability Statement:** Presented data are available upon request to the corresponding author.

**Acknowledgments:** The authors thank the staff of the National Research Center for Geoanalysis and Wuhan Sample Solution Analytical Technology Co., Ltd. for their help with laboratory analyses. We are grateful to the editors and anonymous reviewers for their constructive reviews that significantly improved the quality of this paper.

**Conflicts of Interest:** The authors declare no conflict of interest.

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
