# Peer review of "Petrogenesis and Tectonic Setting of the Madeng Dacite, SW Sanjiang Indosinian Orogen: Evidence from Zircon U-Pb-Hf Isotopes, and Whole-Rock Geochemistry and Sr-Nd Isotopes"

_minerals, doi:10.3390/min12030388_

Round 1

Reviewer 1 Report

This is well-structured and concise study dealing with the origin of magmatic rocks in the Sanjiang orogenic belt, inferred from whole rock geochemistry, U-Pb and Lu-Hf isotopes on zircons, and whole rock Sm- and Nd-isotope data.

I have only minor comments and suggestions, as indicated in the annotated pdf document:

  1. please, rewrite explanations to figure 3
  2. add references to explain sources of unpublished data in Figures 7, 9 and 12

Author Response

1 please, rewrite explanations to figure 3

Response 1: Thanks for your suggestion. We have rewritten the explanations of Figure.3, see line 112 to 120.

2 add references to explain sources of unpublished data in Figures 7, 9 and 12

Response 2: Thanks for your suggestion. we have added the references of published data sources in Figure 7, 9, 12, and 13.

Reviewer 2 Report

This article deals with geochemical and geochronological features of dacites from the Jinshajiang Indosinian belt. As far as I can evaluate them, these data are sound. Nevetheless, I have some general concerns.

  1. First of all the authors use the term of “Sanjiang orogenic belt”. This term is very popular among Chinese geologists. However, it is inaccurate, since there several orogens are juxtaposed due to the Himalayan collision. Sanjiang area is OK as a pure geographic term, but not Sanjiang orogenic belt. Indeed, the right term to be used is “Indosinian orogen”.

The title must be modified accordingly.

  1. The geological setting is poorly introduced. The geological maps are acceptable but cross sections both at the continental crust scale and at the local scale should be also presented, and used to locate the analyzed rocks in the belt bulk architecture.
  2. The proposed interpretation that these rocks derive from the melting of a continental crust is quite reasonable. However, the authors argue that the tectonic setting was an oceanic subduction active from Permian to Late Triassic. Though possible, this view is not in agreement with previous works and thus it must be thoroughly discussed. In the entire Indosinian belt, the deformed rocks are unconformably covered by undeformed red continental beds (molasse-like deposits) of Late Triassic age. Thus subduction was no more active at that time. Moreover, per-aluminous granites dated at ca 244 Ma are involved in the collision (e.g. Zi et al., 2012; Lai et al., 2014; Liu et al, 2014; Faure et al., 2016 and enclosed references). Even if continental crust melting may develop in the upper plate of oceanic subduction belts (e.g. Andes or N. American cordilleras), this is not the unique possibility. Melting of subducted continental crust is another possibility that must be discussed.
  3. Concerning the geodynamic setting, on the basis of general studies on the Indosinian orogen, from Song Ma and Song Chay in N Vietnam to Jinshanjian area in Sichuan, the collision occurred from late Early Triassic to early Middle Triassic. The geochemical analyzes presented in this paper should be discussed in the frame of the entire Indosinian belt with a special attention to rocks derived from crustal melting that are conspicuously present in the entire belt.
  4. In the Intoduction section (line 54), the authors state : “we provided new evidence for the evolution of Paleo-Tethys”. This is a good attempt, unfortunately nothing is presented in the discussion section.

In conclusion, this paper provides important new data that deserve publication. However, a major revision considering the above mentioned points must be done before.

Reviewer 3 Report

In the manuscript “Petrogenesis and Tectonic Setting of the Madeng Dacite, SW 2 Sanjiang Orogenic Belt: Evidence from zircon U-Pb-Hf Iso-3 topes, and Whole-rock Geochemistry and Sr-Nd Isotopes”, the authors present and discuss zircon U-Pb LA-ICP-MS data together with new mineral and whole rock geochemical data of the Madeng dacite.

I must admit, that the authors did not convince me that this study was important and necessary. The introduction should clearly state the problem, (or a gap in knowledge), and the reader should understand why the new data will be able to fill this gap/ solve this problem. The statement that “there was no systematic research” is not sufficient and unfortunately not true (see below).

The text is well-written (with some minor mistakes), and the data support the discussion and conclusions. But the exactly same rocks were already dated by Xin et al. (2018) and Tang et al. (2016, found in Xin et al., 2018). Both publications were cited, but the authors do not state, that the same rock (from almost the same place) was already dated in the past. Additionally, Xin et al (2018) present whole rock geochemical data, Sr-isotopes and Nd-isotopes of the Madeng dacite. At least, these data should be compared explicitely to the new data and differences should be discussed. The only data that were not yet achieved by others are the zircon Hf data, which do not contribute much to the discussion and conclusions.

The uncertainties of the zircon U-Pb LA-ICP-MS data are too small. This has several reasons: 1) the authors give 1σ uncertainties, what is not common in U-Pb dating. The uncertainties should be 2σ (Horstwood et al., 2016). 2) there are no data of a secondary standard. The analysis of a secondary standard allows to assess the true accuracy of the data (Horstwood, 2016) and better estimate the uncertainty. The error given here is an analytical uncertainty, which is often lower than the accuracy. If a secondary standard was dated with the unknowns, the data need to be shown and discussed.

The high MSWD of the zircon U-Pb age of sample MD20-1-2 shows that the data stem from more than one population and should not be combined in one single mean (or Concordia) age. There might be some Pb loss. If I plot the weighted mean diagram of this sample with increasing sample 238U/206Pb ages, I see a plateau at the highest ages (without the outlier) and a tail towards younger ages, what is typically interpreted as Pb loss. Thus, the highest ages should be combined into a weighted mean or Concordia age. By the way, if the highest ages are assumed to represent the crystallization age, this would be within error identical to the ages of Xin et al. (2018), of the same rocks (ca. 247 Ma). This is different for sample MD20-5-2, where I agree with the interpretation of the authors.

I  attach the manuscript, where I marked some smaller mistakes and problems.     

Round 2

Reviewer 2 Report

This revised version has been well improved. The paper is nearly acceptable now. Nevertheless, I suggest to the authors not to use "Sanjianf orogen" because in this area there are two orogenic belts with two distinct ophiolitic sutures.

It is better to use "sanjiang area" or "sanjiang region"